# The Role of ABO Blood Type in Patients with SARS-CoV-2 Infection: A Systematic Review

**DOI:** 10.3390/jcm11113029

**Published:** 2022-05-27

**Authors:** Federico Banchelli, Pierpaolo Negro, Marcello Guido, Roberto D’Amico, Veronica Andrea Fittipaldo, Pierfrancesco Grima, Antonella Zizza

**Affiliations:** 1Department of Medical and Surgical Sciences, University of Modena and Reggio Emilia, 41100 Modena, Italy or banchelli.federico@aou.mo.it (F.B.); roberto.damico@unimore.it (R.D.); 2Unit of Statistical and Methodological Support to Clinical Research, University Hospital of Modena, 41100 Modena, Italy; 3Immunohaematology and Transfusion Medicine Unit, Inter-Company Department of Transfusion Medicine (IDTM) of Local Health Unit (LHU) of Lecce, Vito Fazzi Hospital, 73100 Lecce, Italy; 4Laboratory of Hygiene, Department of Biological and Environmental Sciences and Technologies, University of Salento, 73100 Lecce, Italy; marcello.guido@unisalento.it; 5IRCCS-Istituto di Ricerche Farmacologiche Mario Negri, 20156 Milan, Italy; veronicaandrea.fittipaldo@marionegri.it; 6Infectious Diseases Unit, Vito Fazzi Hospital, 73100 Lecce, Italy; pierfrancescogrima@yahoo.it; 7Institute of Clinical Physiology, National Research Council, 73100 Lecce, Italy; zizza@ifc.cnr.it

**Keywords:** SARS-CoV-2 infection, ABO blood group, coronavirus disease 2019, COVID-19, susceptibility, systematic review

## Abstract

The SARS-CoV-2 infection has caused over 422 million contagions and 5.8 million deaths resulting in a global health crisis. Several studies have investigated the risk factors predisposing to the infection and reported that the host susceptibility can be linked to the ABO blood group, but the current evidence is controversial. We systematically searched for articles in EMBASE, PubMed, and Cochrane library published up to 7 May 2021 to explore the association of the ABO blood group with the susceptibility to SARS-CoV-2 infection. All studies in people undergoing SARS-CoV-2 test controls were included. Odds ratios were obtained in each study and then synthesised by using meta-analysis. Overall, 22 articles were selected and more than 1,200,000 individuals of whom 74,563 resulted positive to SARS-CoV-2 and 1,166,717 resulted negative, were included in the meta-analysis. Overall, 487,985 subjects had blood group A, 151,879 had group B, 52,621 had group AB, and 548,795 had group O. Group O was slightly less associated with infection, as compared to the other three blood groups (OR = 0.91, 95% CI = 0.85–0.99, *p* = 0.02). Conversely, group A was slightly more associated with infection, as compared to the other three groups (OR = 1.06, 95% CI = 1.00–1.13, *p* = 0.04). This meta-analysis shows associations between blood groups and SARS-CoV-2 infection and supports the hypothesis that blood type O may have a slightly lower risk of infection, whereas blood type A may have a slightly higher risk of infection.

## 1. Introduction

In December 2019 atypical pneumonia called coronavirus disease 2019 (COVID-19) was identified for the first time in China. COVID-19 spread around the world resulting in a global health crisis, causing over 422 million contagions and 5.8 million deaths up to February 2022 [1].

The causative agent of this outbreak was the SARS-CoV-2 RNA virus belonging to the βetacoronavirus genus that infects humans by binding to the angiotensin-converting enzyme 2 (ACE2) receptor for cell entry [2]. ACE2 receptors, cellular receptor-binding domain (RBD) of the spike glycoprotein (S protein) of the viral envelope, are expressed in type II alveolar cells of the lung, in endothelial cells in the microcirculation (pulmonary and non-pulmonary), and several human organs, including heart, kidney, liver, and intestine [3].

COVID-19 primarily manifests as an acute respiratory infection, with symptoms such as fever, cough, and shortness of breath. A small percentage of patients also presents gastrointestinal symptoms such as nausea, diarrhoea, and vomiting [4]. COVID-19 is a disease with a broad spectrum of clinical severity which can range from the asymptomatic disease [5] to acute respiratory distress syndrome, with severe lung involvement, endothelial injury, disseminated intravascular coagulopathy, thrombosis, and death due to multiple organ failure [6].

The presence of such different clinical manifestations and the absence of specific symptoms had determined the need for an early laboratory diagnosis of SARS-CoV-2 infection. The detection of viral sequences by real-time reverse-transcription polymerase chain reaction (rRT-PCR) of nasopharyngeal swabs and further confirmation by nuclear acid sequencing is the routine method for diagnosis of infection [7]. Serological tests have also been produced to detect the immunoglobulin M (IgM) and immunoglobulin G (IgG) in individuals in response to SARS-CoV-2 infection and are particularly useful to estimate the level of transmission within a community [8].

By targeting the SARS-CoV-2 spike protein or nucleocapsid protein, rapid results are made available within 30 min at the point of care, with time- and cost-effectiveness advantages, although they have lower specificity and sensitivity compared to corresponding molecular tests [9].

Several studies have investigated the risk factors that predispose to SARS-CoV-2 infection. Advanced age, male gender, and the presence of comorbidities/chronic diseases appeared to be important risk factors for infection and could be associated with severe and even fatal respiratory disease [5]. Additional factors, such as hormonal, environmental, epidemiological, social, and genetic characteristics (including blood type) also appeared to be associated with the susceptibility, severity, and clinical progression of SARS-CoV-2 infection [10,11,12,13].

Regarding the role of the ABO blood group as a risk factor for infection, several studies have shown a different susceptibility to infection amongst the four blood groups. Some authors have shown that blood type O appears to have a protective effect on infection and its progression, whereas blood type A appears to be associated with a higher risk of SARS-CoV-2 infection and severe clinical progression of the disease [3,13,14,15]; however, the existing evidence is still incomplete and firm conclusions were not drawn. Based on these observations, a meta-analysis of the current literature is needed to better clarify the role of the ABO blood type in the onset of SARS-CoV-2 infection.

This review is intended to investigate the role that ABO blood groups play in susceptibility to SARS-CoV-2 infection through a meta-analysis of the results of studies that have evaluated the distribution of blood groups in subjects with SARS-CoV-2 as compared to healthy individuals who tested negative for infection.

## 2. Materials and Methods

### 2.1. Research Strategy and Selection Criteria

This review was conducted according to the Preferred Reporting Items for Systematic reviews and Meta-Analyses (PRISMA) 2020 statement [16] and the Cochrane Handbook for Systematic Reviews of Interventions [17]. The research protocol was registered on PROSPERO (CRD42020196254). Ethical approval was not required for the present review because only published data were used.

The research strategy was performed in MEDLINE (PubMed), EMBASE, and Cochrane Central Register of Controlled Trials, on papers published from 1 January 2020 up to 7 May 2021. Only studies published in the English language were eligible for inclusion. Details on the research strategy are reported in Appendix A.

The list of the identified studies was supplemented with a paper by the same authors of this review which was accepted for publication [18].

Articles that included patients whose ABO blood type was identified and for which the presence or absence of SARS-CoV-2 infection was reported, were evaluated for inclusion.

Studies were excluded if they did not report a certain diagnosis of SARS-CoV-2 infection for subjects whose ABO blood type was known.

No restrictions on age, gender, ethnicity, or health status were applied.

The outcome of the study was the occurrence of SARS-CoV-2 infection, comparing patients with different ABO blood types (A, B, AB, and O).

### 2.2. Selection of Studies and Data Extraction

After the removal of duplicates between databases, the records were independently selected according to pre-specified eligibility criteria by three review authors (PN, MG, AZ) in two separate stages. First, titles and abstracts have been screened for eligibility using the screening tool Rayyan (Available from: https://www.rayyan.ai, accessed on 21 May 2021) and then through the full text. Disagreements on eligibility among the reviewers have been resolved by discussion. Then, data were extracted using a data extraction form by three reviewers (PN, MG, AZ) independently. Disagreements have been solved by discussion among all authors.

Data on study characteristics (authors, country and year of publication, the period in which study was conducted, study design) and participants’ details (number of participants, setting, gender, age, diagnostic test for SARS-CoV-2) were extracted.

### 2.3. Assessment of Risk of Bias in Included Studies

The risk of bias (RoB) in included studies was assessed by using the Newcastle-Ottawa Scale (NOS), recommended by the Cochrane Collaboration [19,20] for quality assessment of observational studies in meta-analyses (Appendix A).

The NOS is an 8-items scale that ranks the RoB in case-control studies (CCS) and cohort studies (CS). Domains of the NOS include selection (CCS and CS), comparability (CCS and CS), exposure (CCS), and outcome (CS).

Each study that was selected for final inclusion in the quantitative synthesis was scored by three reviewers (PN, MG, and AZ) based on its design. Cross-sectional studies were assessed with the NOS for CS. Studies that did not describe their design were assigned by the authors of this review to one of the CCS, CS, and cross-sectional study designs, if possible.

Items of the NOS that are based on the presence of a follow-up period were not applicable. These include the “non-response rate” item for CCS and the “length of follow-up” and “adequacy of follow-up” items for CS. Moreover, items regarding comparability were not assessed as, to our knowledge, confounders that can introduce relevant bias in the results were not identified.

Overall, this modified NOS scale had six items regarding the CCS quality characteristics (case definition adequate, representativeness of the cases, selection of controls, definition of controls, ascertainment of exposure, method of ascertainment) and five items regarding the quality assessment of the CS and cross-sectional studies (representativeness of the exposed cohort, selection of the non-exposed cohort, ascertainment of exposure, outcome of interest not present at the start of the study, and assessment of outcome).

We considered the quality of studies to be high (“low risk of bias”) when all items of the NOS were positively scored, and at “high risk of bias” when at least one item was not positively scored.

The RoB for the included studies was assessed and discussed by all authors.

### 2.4. Data Synthesis

The association between the blood type and the SARS-CoV-2 infection was measured by using the Odds Ratio (OR) for each study and the uncertainty in results was expressed with a 95% confidence interval (CI). Raw results of each study, such as the number of SARS-CoV-2 infected individuals in each ABO blood group, were used to calculate ORs. The principal analysis was set to compare each of the four ABO blood types with the remaining three ones, resulting in four comparisons (O vs. non-O, A vs. non-A, B vs. non-B, AB vs. non-AB). An additional analysis was also performed, considering all six pairwise comparisons between ABO blood groups (A vs. B, A vs. AB, A vs. O, B vs. AB, B vs. O, AB vs. O). All meta-analyses were performed using a random-effects model [21], with subgroup analysis based on the study design (CCS vs. CS or cross-sectional study). The presence of statistical heterogeneity was assessed by visual inspection of the forest plots, as well as by calculating the I2 statistics and their significance. Data synthesis was carried out with R 3.6.3 statistical software (The R Foundation for Statistical Computing, Wien) at *p* < 0.05 significance level.

## 3. Results

### 3.1. Search Outcomes

There were 437 studies identified by our search strategy in May 2021. All of them were evaluated for inclusion by title and abstract screening. Of 314 potentially relevant reports identified after eliminating duplicates, supplemented with one paper of the authors accepted for publication, 23 of them met the inclusion criteria. Five of the 23 included studies were CCS [18,22,23,24,25], and the remaining 18 were CS or cross-sectional studies [26,27,28,29,30,31,32,33,34,35,36,37,38,39,40,41,42,43] as reported by the authors of the papers or deduced from the authors of this review.

A flow diagram describing the selection of studies is reported in Figure 1.

### 3.2. Characteristics of the Included Studies

The characteristics of the included studies that met our inclusion criteria are reported in Table 1.

Six studies were published in 2020 [22,27,28,34,35,39] and 17 studies in 2021 [18,23,24,25,26,29,30,31,32,33,36,37,38,40,41,42,43]. All included studies were carried out in the pre-vaccinal period: between February and July 2020 for studies published in 2020, and until November 2020 for those published in 2021. Eleven of the 23 studies were carried out in Europe [18,24,26,27,28,29,34,36,40,42,43], of which five were in the United Kingdom [24,26,34,36,43], nine studies in America [23,25,30,31,32,35,37,39,41], of which seven were in the USA [23,25,30,31,32,35,39], and only three in Asia (Saudi Arabia, Iraq and Iran) [22,33,38].

In a cohort study, the odds ratios for blood group are reported for participants with at least one positive COVID-19 test compared to participants without a positive test; therefore, it was not possible to obtain data on blood group distribution of COVID-19 positive patients and the control group. This study was not included in the meta-analysis after having tried to gather the data from the authors [34]. Overall, the participants included in the meta-analysis were 1,241,280. The majority (91.1%) of them were assessed in CS or cross-sectional studies (n = 17), whereas only 8.9% of them were assessed in CCS (n = 5). The included studies considered both male and female individuals of all age groups.

Eight studies retrieved data on the general population and four on community volunteers or stem cell or blood donors, while the remaining studies were carried out in hospitalized subjects or in specific groups of subjects (pregnant women, haemodialysis patients, crewmembers, and nephrologists). The differences in the populations under study manifest themselves with a highly heterogeneous risk of SARS-CoV-2 infection across CS or cross-sectional studies (Table 2).

In all studies, the control group was represented by subjects belonging to the same population as the study group, with the exception of the study by Ad’hiah Ah, et al. [22], whose control group was represented by blood donors who tested negative for SARS-CoV-2.

In most studies, the diagnosis of SARS-CoV-2 infection was confirmed by the direct detection of viral RNA using nuclear acid amplification test (NAAT), such as reverse transcription (RT)-PCR, and only in a few studies, the indirect method for detection of antibodies was used (serological assays).

### 3.3. Assessment of Risk of Bias

Assessment of RoB for each included study–using the Newcastle Ottawa Scale (NOS) –was reported in Table 3a for the CCS and in Table 3b for the CS and cross-sectional studies. Overall, three out of five (60%) CCS were at low risk of bias, whereas 16 out of 17 (94.1%) CS or cross-sectional studies were at low risk of bias. One CCS was at high risk of bias for the selection of controls [25] and another one was at high risk of bias for both the selection of controls and the representativeness of the cases [24]. The only CS at high risk of bias was ranked negatively for the “ascertainment of exposure” and the “assessment of outcome” items, as in this study both the ABO blood group and the SARS-CoV-2 infection were self-reported by participants [40].

### 3.4. Association between Blood Groups and SARS-CoV-2 Infection

The study’s participants are a total of more than 1,200,000 individuals of whom 74,563 (6.0%) positive SARS-CoV-2 and 1,166,717 (94.0%) negative SARS-CoV-2 controls.

Overall, 487,985 (39.3%) subjects had blood group A, 151,879 (12.2%) had group B, 52,621 (4.2%) had group AB, and 548,795 (44.2%) had group O.

According to our meta-analysis, group O was slightly less associated with the occurrence of SARS-CoV-2 infection, as compared to the other three blood groups (OR = 0.91, 95% CI = 0.85–0.99, *p* = 0.02) (Figure 2).

This association was similar in CCS (OR = 0.89, 95% CI = 0.72–1.09, *p* = 0.26) and in CS or cross-sectional studies (OR = 0.91, 95% CI = 0.83–1.00, *p* = 0.05), although statistical significance was not observed in both these subgroups. Conversely, group A was slightly more associated with the occurrence of SARS-CoV-2 infection, as compared to the other three blood groups (OR = 1.06, 95% CI = 1.00–1.13, *p* = 0.04) (Figure 3).

A very similar result was observed in CCS (OR = 1.05, 95% CI = 0.93–1.19, *p* = 0.43) and in CS or cross-sectional studies (OR = 1.07, 95% CI = 1.00–1.16, *p* = 0.06). There were instead no differences between subjects with group B vs. subject with non-B group (OR = 1.01, 95% CI = 0.92–1.11, *p* = 0.82) (Figure 4) or between subjects with group AB vs. subject with non-AB group (OR = 1.05, 95% CI = 0.96–1.15, *p* = 0.24) (Figure 5).

The direct comparison of subjects with group O vs. subjects with group A confirmed that the former was less associated with the occurrence of SARS-CoV-2 (OR = 0.91, 95% CI = 0.85–0.98, *p* = 0.01) (Appendix A). All the other pairwise comparisons did not show any relevant difference and are reported in Appendix A.

## 4. Discussion

### 4.1. Summary of Main Results

This systematic review aimed at assessing the role of the ABO blood group in the susceptibility to SARS-CoV-2 infection. Twenty-two observational studies conducted in the pre-vaccinal period, between February and November 2020, in Europe, America, and Asia and comprehending more than 1,200,000 individuals, were included in the meta-analysis. 

Overall, most of the included studies were at low risk of bias.

We found that a potential relationship between the ABO blood group and susceptibility to SARS-CoV-2 infection may exist. Blood type A was statistically more associated with SARS-CoV-2 infection (OR = 1.06), whereas blood group O was less associated with infection (OR = 0.91). In CS or cross-sectional studies, which were the most part of included studies, the results were similar (OR = 1.07 and OR = 0.91, respectively). This, together with the analysis of absolute risks in the single CS and cross-sectional studies, highlights that the differences amongst groups were slight. Several studies have evaluated the correlation between the ABO blood group and the risk of different infections [44,45,46].

ABO blood group was also associated with the development of SARS-CoV infection in a group of healthcare workers in the previous coronavirus outbreak in 2003 [47].

In the case of COVID-19, cell membrane glycoproteins that act as antigenic determinants of the ABO blood types or the iso-agglutinin ABO system could influence the binding of SARS-CoV-2 to ACE2 receptors. Therefore, the increased risk of infection, in people with blood type A may be partly due to the role of substance A in binding SARS-CoV-2 to ACE2 receptors, with a mechanism similar to that of surface heparan sulphate and sialic acid [13,48,49] whereas anti-A natural antibodies in blood type O appear to play a protective role against the viral infection blocking the interaction between ACE2 receptor and the spike protein [50].

### 4.2. Overall Completeness and Applicability of Evidence

EMBASE, MEDLINE (PubMed), and Cochrane Central Register of Controlled Trials were used to search for studies, and data was extracted from the manuscripts. This systematic review included only peer-reviewed articles. We considered only articles that included patients whose ABO blood group was identified and for whom the presence or absence of SARS-CoV-2 infection was detected.

Overall, more than 1,200,000 individuals were included in the meta-analysis, most of whom (960,906 subjects) belonged to the general population, community volunteers, or stem cell or blood donors and not to hospitalized subjects or specific groups of subjects (pregnant women, haemodialysis patients, crewmembers, and nephrologists), therefore more representative of the blood group distribution in the population.

As well, a rigorous methodology was followed to screen and extract the data by three review authors with multidisciplinary expertise.

### 4.3. Potential Biases in the Review Process

Regarding the blood type distribution, there is a high heterogeneity between the different ethnic groups. It is known that a different ABO frequency can affect the likelihood of infection.

Studies from different countries of the world were included in this meta-analysis. Most of the studies come from Europe and Nord America where the groups A and O are the most frequent. No study was identified in Africa and Oceania. We are aware that this could affect the results, although each study included individuals from the same population.

The different study designs also can represent an important bias, but in our meta-analysis very similar results were observed across different study designs.

Furthermore, the vaccination status may affect the results. The vaccination could influence the effects of blood groups on susceptibility to infection. As reported by the authors of selected studies, all articles included in the meta-analysis were carried out between February and November 2020, in the pre-vaccinal period, eliminating the risk of vaccination-related bias.

Only studies published in the English language were included, therefore a language bias may be present, although few journals that publish papers in languages other than English are indexed in MEDLINE or in other international bibliographic databases commonly used for meta-analyses

Finally, we decided to focus exclusively on the role of the blood group in susceptibility to the infection to avoid that other risk factors such as comorbidities, diabetes and hypertension, could represent confounding elements.

Other factors such as age, gender, and body mass index (BMI) can play a role in the SARS-CoV-2 infection. The results of the studies have shown a correlation mainly between these factors and the severity and clinical outcomes of the disease [51,52]. The blood group, on the other hand, appears to be more correlated with infection susceptibility than with severity or mortality, as highlighted in other meta-analyses [53,54,55].

### 4.4. Context for This Review

The results of studies that investigate the association between the ABO blood types and the occurrence of SARS-CoV-2 infection have shown very high heterogeneity. The first studies that showed evidence of a relationship between infection and ABO blood types were conducted in China, and then many hundreds of articles were published in the world with disagreeing evidence.

Some authors have shown that blood type O appears to have a protective effect on infection, whereas blood type A appears to be associated with a higher risk of SARS-CoV-2 infection [13,14,15,56].

Instead, other studies found no significant evidence of the correlation between blood group and occurrence of SARS-CoV-2 infection [18,23,32,36,37], while others have demonstrated only a decreased risk for contracting SARS-CoV-2 infection in individuals with blood group O [27] or an increased risk for subjects with blood group A compared with the other blood groups [33,38].

### 4.5. Agreements and Disagreements with Other Systematic Reviews

To our best knowledge, seven systematic reviews have been published to date on Medline on the association between ABO blood type and SARS-CoV-2 infection: three in 2020 [53,54,57] and four in 2021 [55,58,59,60].

The first three systematic reviews published in 2020 in the early months of the COVID pandemic included very few studies (4 in Pourali F et al. [54] and in Wu BB et al. [53], and 7 in Golinelli D et al. [57]). Pourali F et al. [54] and Wu BB et al. [53] considered unpublished manuscripts available on preprint servers.

The results of these reviews showed a significantly higher risk of infection among individuals with blood type A and decreased odds among those with blood type O.

We obtained similar results but more studies were included in our review, although more selective inclusion criteria were identified; only studies in which both subjects and controls were tested for SARS-CoV-2 were included, excluding those that considered subjects not specifically tested as negative controls and only peer-revised. This approach has allowed us to obtain results of the association between the ABO blood group and susceptibility based on a solid scientific methodology and on a high number of tested subjects.

A significantly lower susceptibility of infection was associated with the O blood group compared to the non-O blood group and a higher risk in A blood type compared with non-A participants also obtained by Gutierrez-Valencia M. et al. with OR, respectively, of 0.88 (0.82–0.94) and 1.08 (1.02–1.15) in a recently published review in which 29 studies updated in May 2021 were included in the analysis of COVID-19 infection prediction [56]. The odds values are very similar to those of our meta-analysis. A comparison of each of the four ABO blood types with the remaining three ones, and an additional analysis considering all six pairwise comparisons between ABO blood groups were performed by both research groups.

An important difference concerns instead the search strategy carried out in Medline, medRxiv, and bioRxiv by Gutierrez-Valencia M et al. [59].

Partially different results were obtained from the other meta-analyses published in 2021. In particular, Liu N et al. [55], found that individuals with group A and B had a substantially higher risk of COVID-19, and individuals with blood group O was not prone to develop the disease. Kabra SM et al. [60], instead observed that the COVID-19 infection rate was higher in people with blood group A > O > B > AB. Therefore, blood group A is linked to a higher risk and the blood type AB to a lower risk of infection. Finally, individuals with group O had a lower infection rate compared to individuals of the non-O group according to the analysis reported in Franchini M et al. [58]. Seventeen studies that fulfilled the inclusion criteria were considered for the analysis of the prevalence of the blood group O versus non-O types. Articles published from January to December 2020 that reported the prevalence of ABO blood groups in SARS-CoV-2 infected and non-infected individuals were included. In addition, subjects not specifically tested for SARS-CoV-2 (e.g., blood donors), collected before the emergence of SARS-CoV-2 [15,61,62] were included in the control groups.

### 4.6. Implications for Practice

Systematic reviews and meta-analyses represent the most reliable scientific tools for evaluating and analysing clinical evidence [63]. The association between blood group and susceptibility to SARS-CoV-2 infection is increasingly evident, although not fully understood.

If blood group O was protective of the infection, this could minimize the clinical course of the infection by ensuring that a person remains asymptomatic and does not undergo diagnostic tests.

However, lower susceptibility to infection does not imply lower transmissibility, as this latter depends also on other conditions, such as clinical and environmental ones.

The results of the meta-analysis could also be used as clinical evidence to better plan health interventions taking into account that different individuals may have different levels of susceptibility to SARS-CoV-2 infection.

In addition, the blood groups of the population could contribute to predicting the course of an outbreak and evaluate strategies to control the spread of SARS-CoV-2 infection also taking into account the distribution of blood groups in different countries.

### 4.7. Implications for Research

As the results of this meta-analysis appear to highlight a protective role of anti-A isoagglutinin against SARS-CoV-2 infection, there is a clear need for titration of isoagglutinin ABO for a better understanding of the role of the blood groups in the infection. Speculatively, higher antibody titres may correspond to a lower risk of infection.

The titre of ABO isoagglutinin declines with age [64,65], further observations also regarding the distribution of the infection across the different ages are needed.

Another important aspect concerns the relationship between the severity of the disease and the ABO blood group; therefore, this investigation could be addressed by the authors in future research.

## Figures and Tables

**Figure 1 jcm-11-03029-f001:**
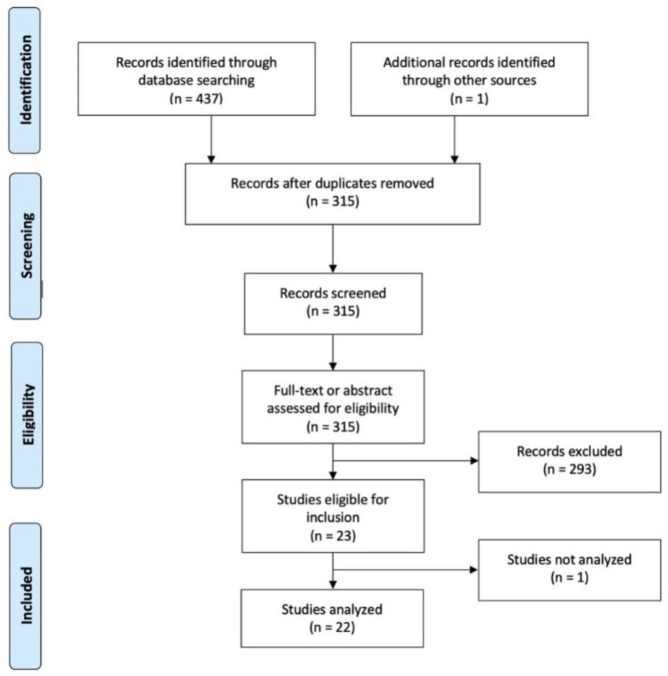
PRISMA 2009 Flow Diagram: ABO blood & SARS-CoV-19.

**Figure 2 jcm-11-03029-f002:**
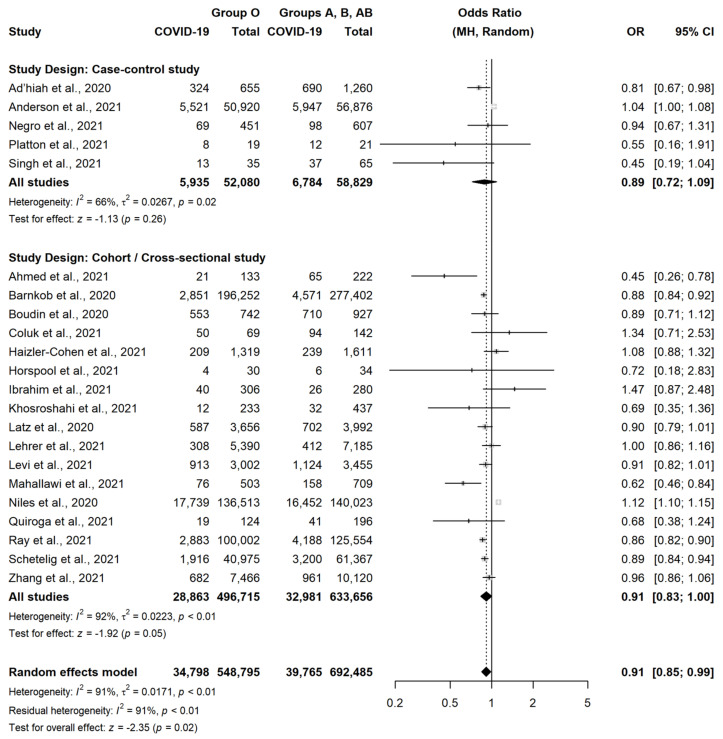
Forest plot for the occurrence of SARS-CoV-2 infection in patients with Blood Group O versus non-O [18,22,23,24,25,26,27,28,29,30,31,32,33,35,36,37,38,39,40,41,42,43].

**Figure 3 jcm-11-03029-f003:**
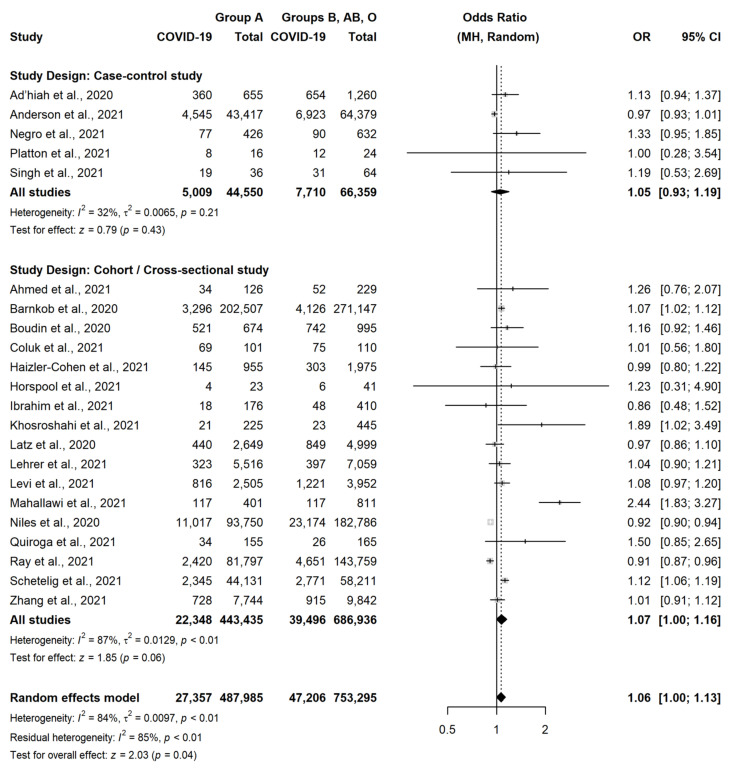
Forest plot for the occurrence of SARS-CoV-2 infection in patients with Blood Group A versus non-A [18,22,23,24,25,26,27,28,29,30,31,32,33,35,36,37,38,39,40,41,42,43].

**Figure 4 jcm-11-03029-f004:**
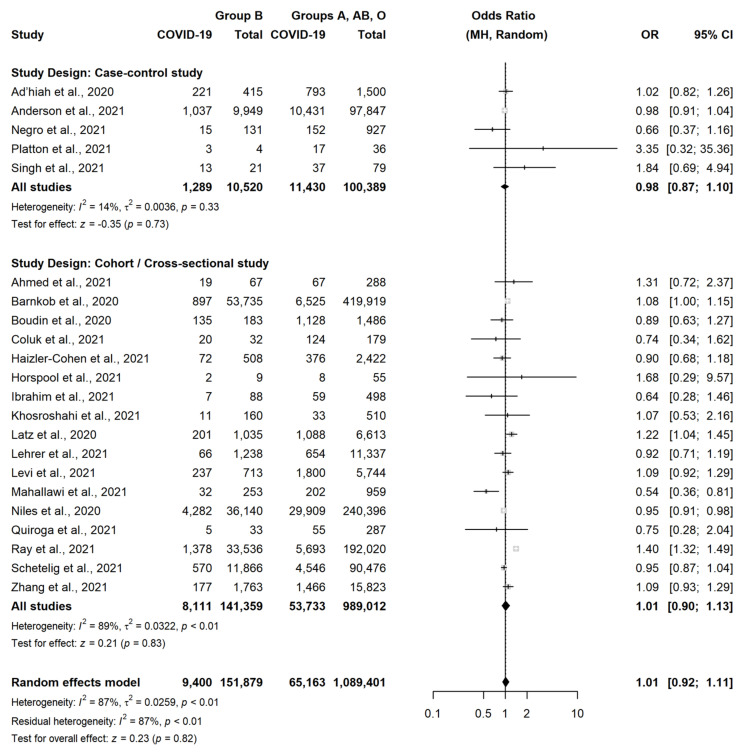
Forest plot for the occurrence of SARS-CoV-2 infection in patients with Blood Group B versus non-B [18,22,23,24,25,26,27,28,29,30,31,32,33,35,36,37,38,39,40,41,42,43].

**Figure 5 jcm-11-03029-f005:**
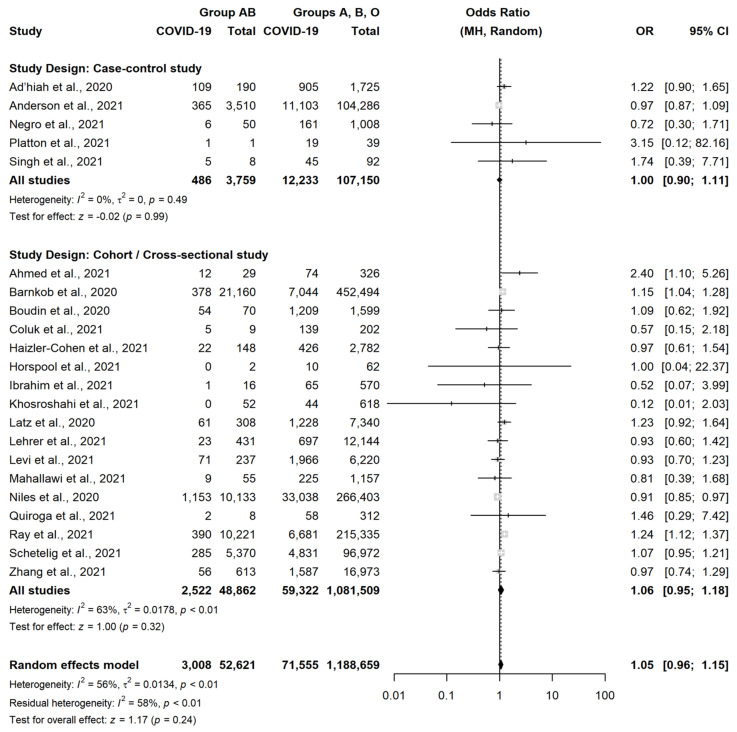
Forest plot for the occurrence of SARS-CoV-2 infection in patients with Blood Group AB versus non-AB [18,22,23,24,25,26,27,28,29,30,31,32,33,35,36,37,38,39,40,41,42,43].

**Table 1 jcm-11-03029-t001:** Characteristics of included study.

Reference	Country	Study Period	Study Design	Participants Included in This Review ^§^	SARS-CoV-2 Positive Patients	SARS-CoV-2 Negative Patients	Gender (M)	Age (Years)	Diagnostic Test
Ad’hiah et al., 2020 [22]	Iraq	31 May to 31 July 2020	Case-control	1915	Hospitalized patients	Blood donors	58.3%	Mean age 39.6	PCR
Ahmed I et al., 2021 [26]	UK	24 April to 6 May 2020	Cohort/Cross-sectional	355	Pregnant women	Pregnant women	0.0%	Mean age 30.8	PCR
Anderson JL et al., 2021 [23]	USA	3 March to 2 November 2020	Case-control	107,796	General population	General population	23.1%	Mean age 42.0	PCR
Barnkob MB et al., 2020 [27]	Denmark	27 February to 30 July 2020	Cohort/Cross-sectional	473,654	General population	General population	29.0%	≥60 years: 35.2% ^1^	PCR
Boudin L et al., 2020 [28]	France	28 February to 13 April 2020	Cohort/Cross-sectional	1669	Crewmembers	Crewmembers	87.0% ^a^	Median age 28.0	PCR
Coluk Y et al., 2021 [29]	Turkey	NR	Cohort/Cross-sectional	211	General population	General population	NR	Subjects > 18 years	PCR
Haizler-Cohen L et al., 2021 [30]	USA	27 May to 28 August 2020	Cohort/Cross-sectional	2930	Pregnant women	Pregnant women	0.0%	Women of reproductive age	Antibody
Horspool A et al., 2021 [31]	USA	NR	Cohort/Cross-sectional	64	Hospitalized patients	Hospitalized patients	56.3%	Range: 15–92	Antibody
Ibrahim SA et al., 2021 [32]	USA	1 March to 31 May 2020	Cohort/Cross-sectional	586	Pregnant women	Pregnant women	0.0%	Women of reproductive age	PCR or antigen
Khosroshahi HT et al., 2021 [33]	Iran	Until 1 July 2020	Cohort/Cross-sectional	670	Haemodialysis patients	Haemodialysis patients	64.5%	Range: 19–88	PCR
Kolin DA et al., 2020 [34]	UK	16 March to 18 May 2020	Cohort/Cross-sectional	4811	General population	General population	53.8% ^1^	Range: 40–69	PCR
Latz CA et al., 2020 [35]	USA	6 March to 16 April 2020	Cohort/Cross-sectional	7648	General population	General population	32.4% ^1^	Subjects > 18 years	PCR or antigen
Lehrer S et al., 2021 [36]	UK	16 March to 26 April 2020	Cohort/Cross-sectional	12,575	Community volunteers	Community volunteers	47.8%	Range: 40–70	PCR
Levi JE et al., 2021 [37]	Brazil	Until 22 June 2020	Cohort/Cross-sectional	6457	General population	General population	NR	NR	PCR and/or antibody
Mahallawi AH et al., 2021 [38]	Saudi Arabia	Mid-May to mid-July, 2020	Cohort/Cross-sectional	1212	Blood donors	Blood donors	100.0%	Range: 18–64	Antibody
Negro P et al., 2021 [18]	Italy	28 February to 23 April 2020	Case-control	1058	General population	General population	46.2%	Range: 1–100	PCR
Niles JK et al., 2020 [39]	USA	March to July 2020	Cohort/Cross-sectional	276,536	Pregnant women at the time of ABO testing	Pregnant women at the time of ABO testing	0.0%	Median age 34.4	RNA NAAT
Platton S et al., 2021 [24]	UK	NR	Case-control	40	Hospitalized patients in critical care unit	Hospitalized patients in critical care unit	82.5%	Range: 22–78	PCR
Quiroga B et al., 2021 [40]	Spain	Until 1 November 2020	Cohort/Cross-sectional	320	Nephrologists	Nephrologists	33.6% ^b^	Mean age 46.0	Self-reported
Ray JG et al., 2021 [41]	Canada	15 January to 30 June 2020	Cohort/Cross-sectional	225,556	General population	General population	29.1%	Mean age 54.0	PCR
Schetelig J et al., 2021 [42]	Germany	January to September 2020	Cohort/Cross-sectional	102,342	Stem cell Donors	Stem cell Donors	29.8% ^c^	Range: 18–61	PCR
Singh N et al., 2021 [25]	USA	1 April to 30 June 2020	Case-control	100	Pregnant women	Pregnant women	0.0%	Range: 17–42	PCR or antigen
Zhang J et al., 2021 [43]	UK	By 24 August 2020	Cohort/Cross-sectional	17,586	Community volunteers	Community volunteers	48.0%	Range: 38–73	PCR

Abbreviations: ^§^ participants tested for COVID-19 with known blood group. ^a^ % on overall participants of the study (1688 individuals). ^b^ % on overall participants of the study (327 individuals). ^c^ % on overall participants of the study. (157,544 individuals). ^1^ % on participants with SARS-CoV-2. NR Not reported.

**Table 2 jcm-11-03029-t002:** Risk of SARS-CoV-2 infection amongst blood groups in cohort studies and in cross-sectional studies.

Study	Blood Group A		Blood Group B		Blood Group AB		Blood Group O		Overall
	Risk	Pos./Tot.	Risk	Pos./Tot.	Risk	Pos./Tot.	Risk	Pos./Tot.	Risk
Ahmed et al., 2021 [26]	27.0%	34/126	28.4%	19/67	41.4%	12/29	15.8%	21/133	24.2%
Barnkob et al., 2020 [27]	1.6%	3296/202,507	1.7%	897/53,735	1.8%	378/21,160	1.5%	2851/196,252	1.6%
Boudin et al., 2020 [28]	77.3%	521/674	73.8%	135/183	77.1%	54/70	74.5%	553/742	75.7%
Coluk et al., 2021 [29]	68.3%	69/101	62.5%	20/32	55.6%	5/9	72.5%	50/69	68.2%
Haizler-Cohen et al., 2021 [30]	15.2%	145/955	14.2%	72/508	14.9%	22/148	15.8%	209/1319	15.3%
Horspool et al., 2021 [31]	17.4%	4/23	22.2%	2/9	0.0%	0/2	13.3%	4/30	15.6%
Ibrahim et al., 2021 [32]	10.2%	18/176	8.0%	7/88	6.3%	1/16	13.1%	40/306	11.3%
Khosroshahi et al., 2021 [33]	9.3%	21/225	6.9%	11/160	0.0%	0/52	5.2%	12/233	6.6%
Latz et al., 2020 [35]	16.6%	440/2649	19.4%	201/1035	19.8%	61/308	16.1%	587/3656	16.9%
Lehrer et al., 2021 [36]	5.9%	323/5516	5.3%	66/1238	5.3%	23/431	5.7%	308/5390	5.7%
Levi et al., 2021 [37]	32.6%	816/2505	33.2%	237/713	30.0%	71/237	30.4%	913/3002	31.5%
Mahallawi et al., 2021 [38]	29.2%	117/401	12.6%	32/253	16.4%	9/55	15.1%	76/503	19.3%
Niles et al., 2020 [39]	11.8%	11,017/93,750	11.8%	4282/36,140	11.4%	1153/10,133	13.0%	17,739/136,513	12.4%
Quiroga et al., 2021 [40]	21.9%	34/155	15.2%	5/33	25.0%	2/8	15.3%	19/124	18.8%
Ray et al., 2021 [41]	3.0%	2420/81,797	4.1%	1378/33,536	3.8%	390/10,221	2.9%	2883/100,002	3.1%
Schetelig et al., 2021 [42]	5.3%	2345/44,131	4.8%	570/11,866	5.3%	285/5370	4.7%	1916/40,975	5.0%
Zhang et al., 2021 [43]	9.4%	728/7744	10.0%	177/1763	9.1%	56/613	9.1%	682/7466	9.3%

Notes: Pos. = individuals positive for SARS-CoV-2; Tot. = total number of individuals tested for SARS-CoV-2.

**Table 3 jcm-11-03029-t003:** (**a**) Quality assessment of Case-Control studies: Newcastle_Ottawa Scale Scores. (**b**) Quality assessment of Cohort or Cross-sectional studies: Newcastle_Ottawa Scale Scores.

(a)						
Study	Selection				Exposure	
	Is the Case Definition Adequate?	Representativeness of the Cases	Selection of Controls	Definition of Controls	Ascertainment of Exposure	Same Method of Ascertainment for Cases and Controls
Ad’hiah AH, et al., 2020 [22]	*	*	*	*	*	*
Anderson JL, et al., 2021 [23]	*	*	*	*	*	*
Negro P, et al., 2021 [18]	*	*	*	*	*	*
Platton S, et al., 2021 [24]	*			*	*	*
Singh N, et al., 2021 [25]	*	*		*	*	*
**(b)**						
**Study**	**Selection**					**Outcome**
	**Representativeness of the exposed cohort**	**Selection of the non-exposed cohort**	**Ascertainment of exposure**	**Demonstration that outcome of interest was not present at start of study**		**Assessment of outcome**
Ahmed I, et al., 2021 [26]	*	*	*	*		*
Barnkob MB, et al., 2020 [27]	*	*	*	*		*
Boudin L, et al., 2020 [28]	*	*	*	*		*
Coluk Y, et al., 2021 [29]	*	*	*	*		*
Haizler-Cohen L, et al., 2021 [30]	*	*	*	*		*
Horspool A, et al., 2021 [31]	*	*	*	*		*
Ibrahim SA, et al., 2021 [32]	*	*	*	*		*
Khosroshahi HT, et al., 2021 [33]	*	*	*	*		*
Latz CA, et al., 2020 [35]	*	*	*	*		*
Lehrer S, et al., 2021 [36]	*	*	*	*		*
Levi JE, et al., 2021 [37]	*	*	*	*		*
Mahallawi AH, et al., 2021 [38]	*	*	*	*		*
Niles JK, et al., 2021 [39]	*	*	*	*		*
Quiroga B, et al., 2021 [40]	*	*		*		
Ray JG, et al., 2021 [41]	*	*	*	*		*
Schetelig J, et al., 2021 [42]	*	*	*	*		*
Zhang J, et al., 2021 [43]	*	*	*	*		*

* The item is respected. The items “Comparability of cases and controls on the basis of the design or analysis” and “Non-Response rate” are not applicable. The items “Comparability of cohorts on the basis of the design or analysis”, “Was follow-up long enough for outcomes to occur” and “Adequacy of follow up” are not applicable.

## Data Availability

Not applicable.

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
