# Peer review of "The Role of ABO Blood Type in Patients with SARS-CoV-2 Infection: A Systematic Review"

_jcm, 2022, doi:10.3390/jcm11113029_

Round 1
Reviewer 1 Report
I think that the authors need to clarify "less susceptible" does not imply "less transmissible". Also, age, sex and BMI are also important factors to consider if available. The authors need to justify the significance of blood type beyond those demographics data.
Author Response
We thank the Reviewer for the revision of the manuscript and we have provided the clarifications as suggested.
We underlined in the discussion that a minor susceptibility does not imply lower transmissibility.
We also highlighted in the discussion how factors such as age, gender, and BMI can play a role in the SARS-CoV-2 infection. However, to correctly estimate how these factors can modify the association between blood type and SARS-CoV-2 infection, individual patient data from all the included studies are needed. In the present study, we decided to use only aggregated published data, as reported by the authors.
We also justified the significance of blood type beyond those demographics data, as the blood group appears to be more correlated with infection susceptibility than with severity or mortality, as highlighted in other meta-analyses.
Reviewer 2 Report
The authors systematically reviewed several studies investigating associations between the ABO blood type and SASR-COV-2 infection. They performed a meta-analysis of data reported in 22 articles and more than 1,200,000 individuals involved. The aim of the study was to assess the role of ABO in the susceptibility to SARS-COV-2 infection.
The review was conducted according to guidelines for systematic reviews and meta-analysis (PRISMA, Cochrane), their research strategy was described in detail, comprehensive literature search in relevant databases were performed. The search terms were provided, the selection of studies and data extraction was described.
All the studies included were carried out in the pre-vaccinal period. Only peer reviewed articles were included. Quality assessment of the different studies included and reported. Suitable analysis including subgroup analysis were performed. The presence of statistical heterogeneity was assessed.
The risk of bias in included studies was assessed and discussed.
Possible limitations of the study including the heterogeneity of ABO blood type distribution between different ethnic groups were discussed.
The results are exact and support the hypothesis that ABO blood type O may have a slightly lower risk of infection.
Minor issue:
-Since only studies published in English language were included, a possible language bias may be present and should be mentioned in the discussion.
-The reasons for the studies excluded should be reported in the materials and methods section.
Author Response
We thank the reviewer for the revision and for the comments which allowed us to highlight a possible language bias and clarify the study exclusion criteria.
We have therefore mentioned in the discussion section a possible bias due to the inclusion of studies published only in the English language.
Furthermore, as suggested by the reviewer, we have reported in the materials and methods section the reasons for the studies being excluded.